# Water Quality Assessment: A Quali-Quantitative Method for Evaluation of Environmental Pressures Potentially Impacting on Groundwater, Developed under the M.I.N.O.Re. Project

**DOI:** 10.3390/ijerph17061835

**Published:** 2020-03-12

**Authors:** Giovanni De Filippis, Prisco Piscitelli, Idelberto Francesco Castorini, Anna Maria Raho, Adele Idolo, Nicola Ungaro, Filomena Lacarbonara, Erminia Sgaramella, Vito Laghezza, Donatella Chionna, Alberto Fedele, Biagio Galante, Raffaele Stasi, Giuseppe Maggiotto, Emanuele Rizzo, Fabio Rocco Nocita, Giovanni Imbriani, Francesca Serio, Paolo Sansò, Alessandro Miani, Antonella De Donno, Domenico Gramegna, Vincenzo Campanaro, Salvatore Francioso, Roberto Bucci, Roberto Carlà, Rodolfo Rollo, Deborah V. Chapman, Vito Bruno

**Affiliations:** 1Local Health Authority ASL Lecce, 73100 Lecce, Italy; giov.defilippis@gmail.com (G.D.F.); francesco.castorini@gmail.com (I.F.C.); annamaria.raho@gmail.com (A.M.R.); sisp@ausl.le.it (A.F.); biagiogalante@gmail.com (B.G.); dr.raffaele-stasi@libero.it (R.S.); giuseppe.maggiotto@gmail.com (G.M.); emanuele.rizzo@email.com (E.R.); nocita.fabio@libero.it (F.R.N.); dirsan@ausl.le.it (R.C.); dirgen@ausl.le.it (R.R.); 2Department of Biological and Environmental Sciences and Technologies (Di.S.Te.B.A.), University of Salento, 73100 Lecce, Italy; adeleidolo73@gmail.com (A.I.); giovanni.imbriani@unisalento.it (G.I.); francesca.serio@unisalento.it (F.S.); paolo.sanso@unisalento.it (P.S.); antonella.dedonno@unisalento.it (A.D.D.); 3Regional Agency for Environmental Protection ARPA Puglia, 70126 Bari, Italy; n.ungaro@arpa.puglia.it (N.U.); m.lacarbonara@arpa.puglia.it (F.L.); e.sgaramella@arpa.puglia.it (E.S.); v.laghezza@arpa.puglia.it (V.L.); d.chionna@arpa.puglia.it (D.C.); d.gramegna@arpa.puglia.it (D.G.); v.campanaro@arpa.puglia.it (V.C.); r.bucci@arpa.puglia.it (R.B.); dg@arpa.puglia.it (V.B.); 4Italian Society of Environmental Medicine (SIMA), 20149 Milan, Italy; alessandro.miani@gmail.com; 5Department of Environmental Science and Policy (ESP), University of Milan, 20100 Milan, Italy; 6Environmental Division, Province of Lecce, 73100 Lecce, Italy; sfrancioso@provincia.le.it; 7UNEP GEMS/Water Capacity Development Centre, Environmental Research Institute, University College Cork, T23 XE10 Cork, Ireland; d.chapman@ucc.ie

**Keywords:** water quality assessment, groundwater, water monitoring, georeferentiation

## Abstract

*Background:* At global level, the vulnerability of aquifers is deteriorating at an alarming rate due to environmental pollution and intensive human activities. In this context, Local Health Authority ASL Lecce has launched the M.I.N.O.Re. (Not Compulsory Water Monitoring Activities at Regional level) project, in order to assess the vulnerability of the aquifer in Salento area (Puglia Region) by performing several non-compulsory analyses on groundwater samples. This first paper describes the quali-quantitative approach adopted under the M.I.N.O.Re. project for the assessment of environmental pressures suffered by groundwater and determines the number of wells to be monitored in specific sampling areas on the basis of the local potential contamination and vulnerability of the aquifer. *Methods:* We created a map of the entire Lecce province, interpolating it with a grid that led to the subdivision of the study area in 32 quadrangular blocks measuring 10 km × 10 km. Based on current hydrogeological knowledge and institutional data, we used GIS techniques to represent on these 32 blocks the 12 different layers corresponding to the main anthropic or environmental type of pressures potentially impacting on the aquifer. To each kind of pressure, a score from 0 to 1 was attributed on the basis of the potential impact on groundwater. A total score was assigned to each of the 32 blocks. A higher number of wells was selected to be monitored in those blocks presenting higher risk scores for possible groundwater contamination due to anthropic/environmental pressures. *Results:* The range of total scores varied from 2.4 to 42.5. On the basis of total scores, the 10 km × 10 km blocks were divided into four classes of environmental pressure (1st class: from 0,1 to 10,00; 2nd class: from 10,01 to 20,00; 3rd class: from 20,1 to 30,00; 4th class: from 30,01 to 42,50). There were 11 areas in the 1st class, 9 areas in the 2nd class, 8 areas in the 3rd class and 4 areas in the 4th class. We assigned 1 monitoring well in 1st class areas, 2 monitoring wells in 2nd class areas, 3 monitoring wells in 3rd class areas and 4 monitoring wells in 4th class areas. *Conclusion:* The methodology developed under the M.I.N.O.Re. project could represent a useful model to be used in other areas to assess the environmental pressures suffered by aquifers and the quality of the groundwater.

## 1. Introduction

Drinking water represents only 0.1% of all the water on the Earth. In addition to water originating from the mountains (natural springs, torrents, rivers), groundwater is the most precious natural resource of water supply for human health and activities and it is undoubtedly one of the greatest providers of life support functions [1]. About 75% of European Union (EU) residents depend on groundwater for their water supply [2], but uses and applications are often related to groundwater composition, which is increasingly influenced by human activities.

Indeed, the initial water composition is primarily related to the water recharge (during the phase of infiltration from the surface), groundwater flow patterns and the general characteristics of aquifer [3]. Changes in water composition may occur through natural processes or as a consequence of human activities on the basis of soil conditions and land use (i.e., evapotranspiration and dissolution of fertilizers). The vulnerability of aquifers is globally deteriorating at an alarming rate due to environmental pollution and intensive human activities [4]. For example, groundwater quality could be impacted by the deposition of air pollutants or chemicals and biologic wastes directly released in soil or water: leaking underground storage tanks, dilapidated wastewater conveyance systems, poorly managed septic systems, unregulated land discharge of various organic and inorganic or toxic wastes, radioactive contaminants from nuclear plants and others. Much of the time, surface run-off originates from areas that have received application of a variety of chemicals such as pesticides, and it is known that herbicides often act as a sink for groundwater pollution. In particular, the Salento area (a subregion in Southern Apulia (Italy), is affected by two types of human-related pollution: salt contamination, which is spreading over large portions of land, with consequent reduced availability of good quality water [5]; and chemical-physical or biological pollution, mainly originating from agricultural activities [6].

The Salento area is characterized by intensive agricultural activity which requires large amounts of irrigation, with groundwater covering 75% of the irrigation demand for the local population due to the absence of rivers or springs [7,8]. The publication of the Water Quality Report issued in 2016 by the National Agency for Environmental Protection (ISPRA)—reporting 2013–2014 data—has highlighted the need for improving the knowledge about groundwater quality in the Apulia Region, referring to use of pesticides in agriculture, compared with other Northern Italian Regions, where information about the higher numbers and types of parameters is already available [9]. The fundamental role of groundwater in the Salento area and the scarcity of data, calls for monitoring activities investigating of the evolution of groundwater quality and its vulnerability, including a specific assessment of anthropic activities impact. Furthermore, for effective pollution control and water resource management, it is necessary to identify potential pollution sources and their quantitative contributions [10].

On this basis, the Local Health Authority of Lecce (ASL Lecce) has decided to run the M.I.N.O.Re. Project (Not compulsory water monitoring activities at regional level), to integrate the currently existing monitoring systems of groundwater and to ultimately protect public health. After the approval of Apulia Regional Council, the M.I.N.O.Re. Project was started in strong cooperation with the Regional Agency for Environmental Protection (ARPA Puglia), Apulian Aqueduct (AQP), University of Salento, Province of Lecce, and other regional and municipal institutions. The main goal of the M.I.N.O.Re. Project is to provide a decisive contribution to improve knowledge about the quality of Salento groundwater by increasing the number and type of analytes currently tested in water samples. These additional analyses integrate those carried out in a compulsory way according to European Water Framework Directive (2000/60/EC), National and Regional laws. 

More specifically, the M.I.N.O.Re. Project consists of six different specific objectives. The first one is devoted to broadening the spectrum of parameters already monitored in drinking water samples from all of the 104 wells used by AQP to deliver potable water and from about 100 fountains (one from every town of Lecce province). The second objective is aimed at monitoring a sample of the overall 30,000 officially authorized wells in the entire Lecce province for agricultural or uses other than drinking, and at increasing the number of monitored vegetables and farm products (meats, honey, milk, eggs). The other specific objectives of the project concern the production of an updated report on Health and Environment in province of Lecce, an experimental integrated health risk assessment (hydric footprint), epidemiologic surveillance and dissemination actions directed to schoolchildren (educational projects) and general population (television, radio and social media) to foster a more responsible attitude towards water use and consumption. All the data coming from the M.I.N.O.Re. project will be made available to the GEMS/Water (Global Environment Monitoring System for freshwater), water quality database, GEMStat, established at global level by the United Nations Environment Programme (UNEP) with the aim of collecting data from all over the world about the quality of groundwater, rivers and lakes that may impact on human health and environment.

The second objective of the M.I.N.O.Re. project is discussed in this article. It includes a particular challenge consisting of the selection of about 70 wells (established on the basis of the project budget) that should be adequately representative of the quality of the aquifer, giving more weight to areas that have more contamination sources. The existing groundwater sampling methodologies are based only on the assessment of the intrinsic groundwater vulnerability to pollution, without considering the pressures present in the area [11,12,13]. Index methods are very popular in vulnerability assessment, where classification of aquifer area is done based on geological and hydrogeological factors [14]. Instead, we have adopted a specific methodology, according to which we define the localization of the wells to be monitored on the basis of the proximity to environmental pressures and human activities (i.e., industry, legal/illegal waste disposal sites, etc.) resulting in potential contamination of groundwater, while taking into account hydrogeological variables (i.e., flow direction, presence of sinkholes, etc.). Research on groundwater contamination in this area and associated human health risks is rather limited [15,16]. Therefore, in this paper we describe the integrated approach used under M.I.N.O.Re. project for establishing a groundwater sampling plan in the Salento area (Apulia Region, Southern Italy), that included individuation of the main anthropic or environmental pressures, their weight and geological mapping. In particular, we have designed a quali-quantitative method to identify the most vulnerable areas to pollutant load distribution in groundwater, on the basis of main anthropic and natural pressures on the environment.

## 2. Materials and Methods 

### 2.1. Study Area

The Salento peninsula is a typical Mediterranean basin, which is bounded to the east by the Adriatic sea, to the west by the Ionian sea and to the north by the “Murge” hill (Figure 1).

For its karst nature, this territory has few surface water resources but, on the contrary, has remarkable groundwater resources for the presence of a multi-layered aquifer characterized by the presence of two distinct systems: a shallow multilevel aquifer, which occupies only 35% of the territory and an extensive aquifer intensively exploited as drinking water [17] and irrigation water [18], and which is constantly threatened by the intrusion of salt water [19].

From a hydrogeological point of view, the study area is characterized by calcareous outcrops highly permeable to water due to the large amount of cracks and dolines associated with karst phenomena [20]. In fact, the most intense precipitations of the cold seasons are rapidly absorbed by the soil and infiltrate directly into the deep aquifer, which rests on a base of sea water into hydrodynamic balance [21,22].

### 2.2. The Modeling Approach

To define the localization of the wells to be monitored, within a maximum pre-defined number of about 70 wells, we decided to start with the definition of a complete list of local environmental pressures and human activities (i.e., industries, legal/illegal waste disposal sites, etc.) potentially impacting on groundwater based on the most updated knowledge and recent scientific literature [23,24,25,26,27]. We have also taken into account the major hydrogeological variables (i.e., flow direction; the presence of cracks, caves, sinkholes, etc.) that can potentially increase the vulnerability of groundwater in terms of pollutants’ load distribution, infiltration or diffusion [19,28,29].

Therefore, in order to determine the main anthropic and natural pressures on the aquifer, we generated a map of the entire province of Lecce using the geographic information system (GIS) and interpolated it with a grid that led to the subdivision of the study area into quadrangular blocks measuring 10 km × 10 km. Along the coast, due to the presence of the sea, some blocks presented smaller dimensions, so that they were incorporated into the nearest ones. As a result of this subdivision, the study area was divided into 32 blocks (Figure 2). The geographical information system (GIS) was used in this work as a tool that was able to manage large set of data and complex modeling environments [30].

Once the subdivision was obtained, the following 12 indicators of anthropic/environmental pressure on groundwater were identified and represented on the map of Salento as different layers by using GIS techniques:Urban Waste Water discharges: the waste water treatment plants, after the purification process, obtained as a result water that can be released on the ground; this information layer indicates the location of waste water discharges;Industrial Plants (Integrated Environmental Authorization): these are types of production installations that can produce significant environmental damage and that must therefore be subject to Integrated Pollution Prevention and Control dictated by the European Union starting from 1996 (IPPC n. 96/61/EC) [31];Potentially Contaminated Sites (Legislative Decree 152/2006 art. 240) [32]: these are areas to be remediated in which an overt environmental impact is recognized (example: a former landfill);Companies authorized for Waste storage and management (Legislative Decree 152/2006 art. 208) [32]: companies that require the storage of waste for the management of its production cycle, or companies that hold waste and store it;Companies authorized to emit certain atmospheric pollutants (Legislative Decree 152/2006 art. 269) [32]: companies that foresee emissions of environmental contaminants into the atmosphere for industrial activities;PCB Waste treatment and management Plants: companies authorized for the storage and dehalogenation of polychlorinated biphenyls in the province of Lecce;Livestock farm: animal farms can represent an important and dangerous source of biological pollution for groundwater;Active quarries: the extractive areas are quarries of limestone used for building;Urban solid waste landfill: these are areas identified by the competent authorities in which the solid urban waste is definitively stored;Major accident hazard industrial activities: these are plants subject to “major accident hazards”. This term, in the context of environmental legislation, indicates the probability that, due to uncontrolled phenomena, a fire or an explosion could give rise to health hazards arising from certain substances dangerous for human health and/or the environment;Disused and abandoned quarries: former mining areas that have been closed since the removal of the limestone. The removal of the rock has weakened the natural filtering capacity that purifies the washing water before reaching the deep aquifer;Sinkholes: a sinkhole, in geology, is the point on a karst surface where water penetrates or sinks into the underground. The informative layer of the sinkholes was considered as a natural pressure, because from a geological point of view, the Salento peninsula is a karst territory consisting of a series of carbonate formations [33]. Aquifers and karst environments are highly vulnerable to contamination and anthropogenic changes. The vulnerability of karst aquifers to contamination is due to particular features such as thin soils, reloading points in sinkholes and swallowing holes [34].

According to information provided by the local health authority ASL Lecce and by the regional environmental agency ARPA Puglia, the 12 layers described above were created using GIS software to differentiate potential sources of risk for groundwater contamination.

After entering the information levels in the GIS database, the “weight” that was assigned to each individual information layer was established on the basis of prior knowledge and scientific literature [23]. Then rating values ranging from 0 to 1 were given depending on the contribution of each anthropic/environmental pressure to the pollution of groundwater. Obviously, each layer had a different weight depending on its relevance in terms of possible threats and the pressure on the aquifer (Table 1).

Subsequently, we computed total scores for each of the 32 blocks measuring 10 km × 10 km presented in Figure 1. 

For each block, into which the entire province of Lecce was divided, the total score was calculated as the weighted sum of the twelve layers using the following equation in a GIS tool:(1)$$\overset{12}{\underset{i=1}{\sum }}n_{i}w_{i}$$
(*i* = anthropic/environmental pressure; *n* = number of points of pressure *i*; *w* = weight attributed to pressure *i*).

As a result, we were able to determine the potential pressure on groundwater for each block of the study area. Those blocks characterized by higher total scores were assumed to be areas at greater risk of groundwater vulnerability, consequently presenting greater risk of potential groundwater contamination due to anthropogenic or environmental pressures. On this basis, we identified a higher number of wells to be monitored in the areas with higher total scores compared to areas with lower total scores.

## 3. Results

Thanks to information provided by the local health authority ASL Lecce and the regional environmental agency ARPA Puglia, the total number of the analyzed anthropic/environmental pressures that persist on the Salento, and their relative total score, were obtained (Table 2).

Figure 3 presents the total score summarizing the anthropic/environmental pressures for each block measuring 10 km × 10 km into which the entire province of Lecce was divided. The total scores presented a range varying from 2.4 to 42.5. On the basis of total scores, the 10 km × 10 km blocks were divided into four classes (1st class: from 0,1 to 10,0; 2nd class: from 10,1 to 20,0; 3rd class: from 20,1 to 30,0; 4th class: more than 30,0). Out a total of 32 blocks, there were four areas corresponding to the highest risk (4th class), eight areas in 3rd class, nine areas in 2nd class and eleven blocks in 1st class.

In Figure 4, we display the graphical elaboration of the different specific anthropic and natural environmental pressures that contributed to the total rating for each block.

We assigned an increasing number of wells to be monitored according to the risk class of each block. Therefore, the highest number of monitoring wells was attributed to the blocks belonging to the 4th class (higher vulnerability and consequent higher risk of potential groundwater contamination). Specifically, as presented in Figure 5, we set one monitoring well in 1st class areas, two monitoring wells in the 2nd class areas, three monitoring wells in 3rd class areas, and four monitoring wells in the 4th class areas for a total of 69 monitoring wells (as required by the second specific objective of M.I.N.O.Re. project).

## 4. Discussion

Due to its karstic nature, the Salento Peninsula is characterized by scarce availability of surface water resources. On the other hand, important volumes of groundwater are hosted in the deep karst aquifer, which represents a strategic resource for the socioeconomic aspects of this Mediterranean region [28,35]; it is very important to emphasize that karstic aquifers and environments are highly vulnerable to anthropogenic contamination.

The results of the M.I.N.O.Re. project could contribute to reduce the knowledge gaps on the Salento aquifer, which is vulnerable to surface pollutants due to the karst nature of the subsoil, with particular reference to pesticides used in agriculture. In this regard, it was ascertained that Puglia is in seventh place in Italy for consumption of plant protection products, with the province of Lecce among the top places at the regional level [9].

Based on the elaboration of the different specific anthropic and natural environmental pressures that persist on the Salento, the result was that the greater risks with respect to the quality of the groundwater are mainly due to the presence of: sinkholes; Companies authorized to emit certain atmospheric pollutants; livestock farm and Companies authorized to waste storage and management.

The presence of 28 potentially contaminated sites awaiting remediation, and 69 Companies authorized to waste storage and management, threatens the qualitative state of the aquifer; these sites increase significantly if they include those of abandonment, discarded fuel distributors and car wrecks or deriving from the suspected illegal burying of toxic substances under investigation by the Judiciary, of which the PCB case of the former Burgesi landfill is only the latest and most striking. By the way, it should also be noted that, while the other Apulian provinces were only interested in temporary PCB storage sites, in the Province of Lecce there are 7 plants with dehalogenation or heat treatment. Taking into account this particular situation, it must be pointed out that an additional number of 5 wells have been sampled and tested for dioxins and PCBs compounds in the blocks 26, 27, 29 and 30, corresponding to the area of Burgesi landfill.

The province of Lecce receives about one thousand requests for the authorization of wells for different uses each year (with an estimated water consumption of 4 cubic meters per second, sometimes improperly used for human use). This problem concerns health authorities in the medium-term perspective because of a progressive and irreparable salinization of the deep-water waters that represent the primary source of drinking water of the Salento region.

Currently, it the existence of approximately 80,000 private wells into the Province of Lecce (including the 30,000 officially authorized) building has been estimated. For this reason, since 2016, AQP could no longer dispose the sewage sludge of civil waste water in the Salento agricultural land. Indeed, the National Agriculture Action Plan also contains—among other things—“strategic guidelines for the definition and implementation of the program of measures relating to the agricultural sector in the second cycle of management plans”, not only dictating directives on the correct use of sludge, but also the efficient and sustainable use of water in agriculture, and the measures for controlling groundwater abstractions and protecting pollution.

The Local Health Authority has elaborated “Guidelines for the correct use of water resources” and it is planning a permanent observatory with the mayors and active citizens’ associations—preparing information programs for citizens and farmers on the correct use and protection of water resources. New indications to the political decision-maker may derive from these activities in terms of adopting more stringent rules to protect groundwater starting from subordinating the renewal of authorizations of new wells to more restrictive criteria and providing the inclusion of more detailed analysis that also include some heavy metals and pesticides. At the same time, the Apulia Region is defining the Water Resources Protection Plan with the aim of regulating and specifying the sustainable use of water resources and groundwater reserves.

A total of one thousand schoolchildren have already been trained in specific campaigns ruled by Local health authority ASL Lecce to foster a correct use of water and two thousand additional students will be reached in 2020. Media campaigns have started on television by broadcasting mini-spots aimed at increasing the knowledge among the general population about the need for preserving the unique natural source of groundwater present in Salento area.

## 5. Conclusions

Groundwater is a critical resource that is increasingly being threatened due to anthropogenic and natural activities. Its management includes a wide range of activities including the prevention of groundwater contamination and a more conscious use of this resource. Vulnerability and pollution risk assessments are the very first important steps in generating useful information for devising strategies aimed at groundwater protection from contamination. Delineating vulnerable zones helps water resource managers to divert groundwater development activities to other safer areas and hence can minimize the cost of water treatment. In this view, the research activities presented in this paper represent an advancement with respect to the previous works and lay the basis for further scientific investigations on this topic, aimed at characterizing the hydrogeological equilibrium of the deep aquifer. Therefore, a quali-quantitative approach was adopted for the assessment of environmental pressures on groundwater as a rationale for determining the monitoring wells representative of the vulnerability of the aquifer in the Salento area. From a qualitative point of view, different types of pressures potentially impacting on the aquifer were chosen, and to each kind of pressure a score was attributed, on the basis of the potential impact on groundwater, in order to define them quantitatively. This manuscript presents a clear and practical approach for establishing a groundwater sampling plan. Therefore, the paper could be a useful resource for public authorities with a mandate to monitor groundwater quality. In conclusion, the methodology presented in this paper is relevant for environmental monitoring applications, and—if integrated with specific local hydrogeological analyses—could represent a useful model that could be used in other areas to assess the status of the available water resources and their quality.

## Figures and Tables

**Figure 1 ijerph-17-01835-f001:**
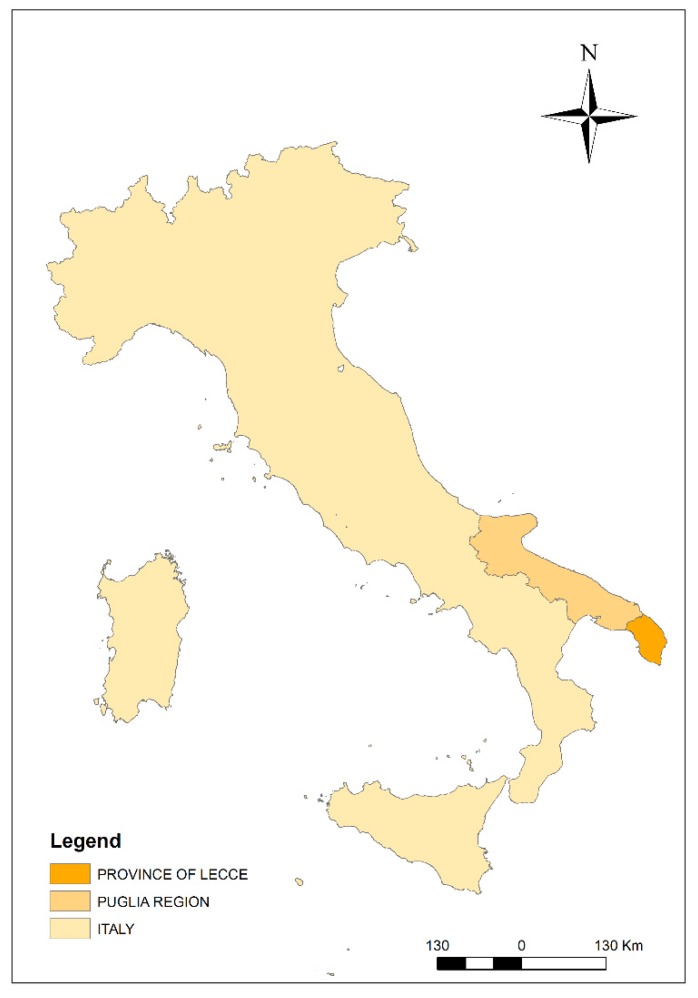
Study area: the Salento peninsula (Apulia Region, Southern Italy).

**Figure 2 ijerph-17-01835-f002:**
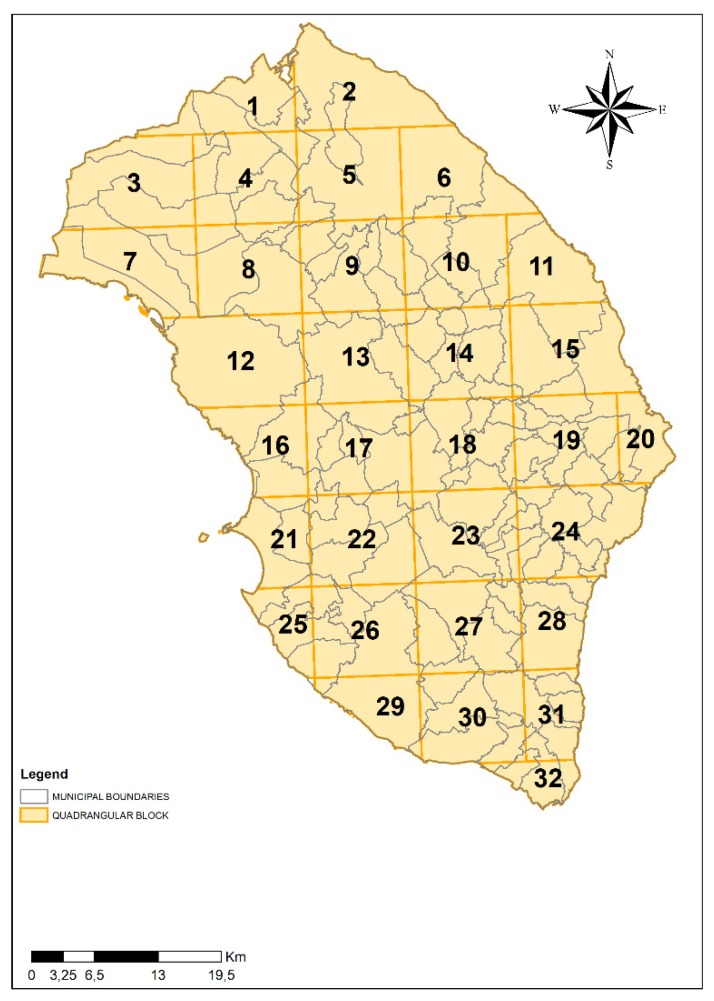
Subdivision of the study area in 32 quadrangular blocks measuring 10 km × 10 km.

**Figure 3 ijerph-17-01835-f003:**
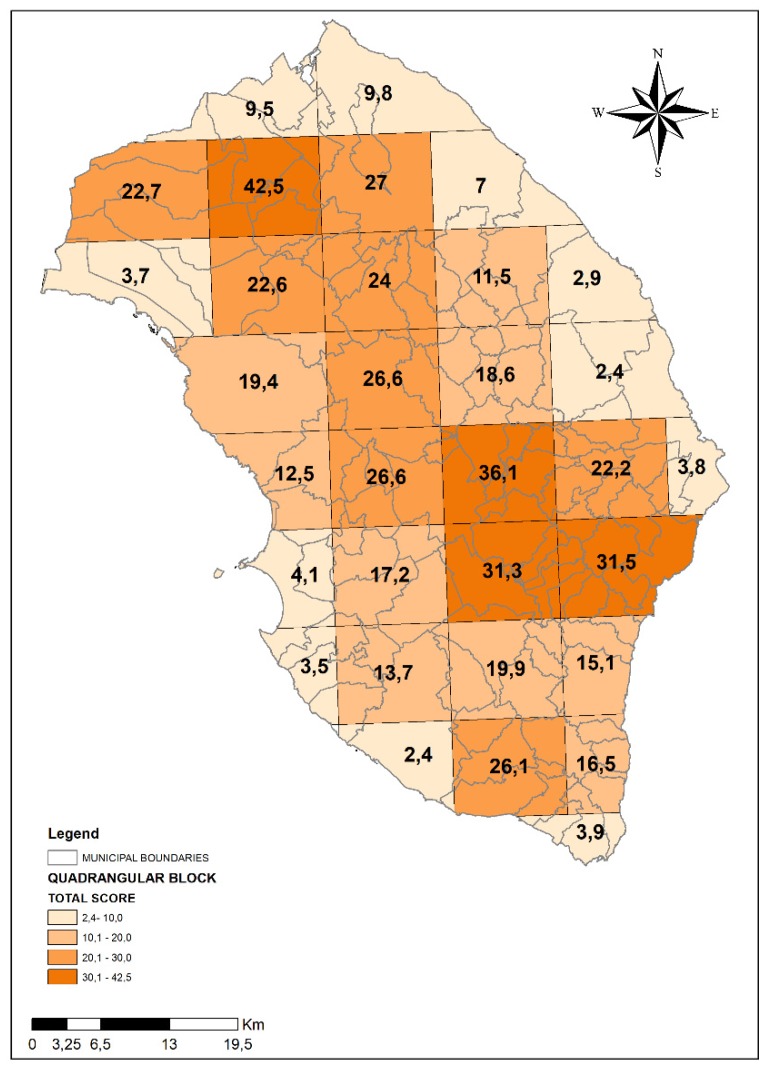
Map of Lecce province divided into four different classes according to anthropic/environmental pressures that can potentially impact on groundwater (the four classes are represented with different intensity of colors).

**Figure 4 ijerph-17-01835-f004:**
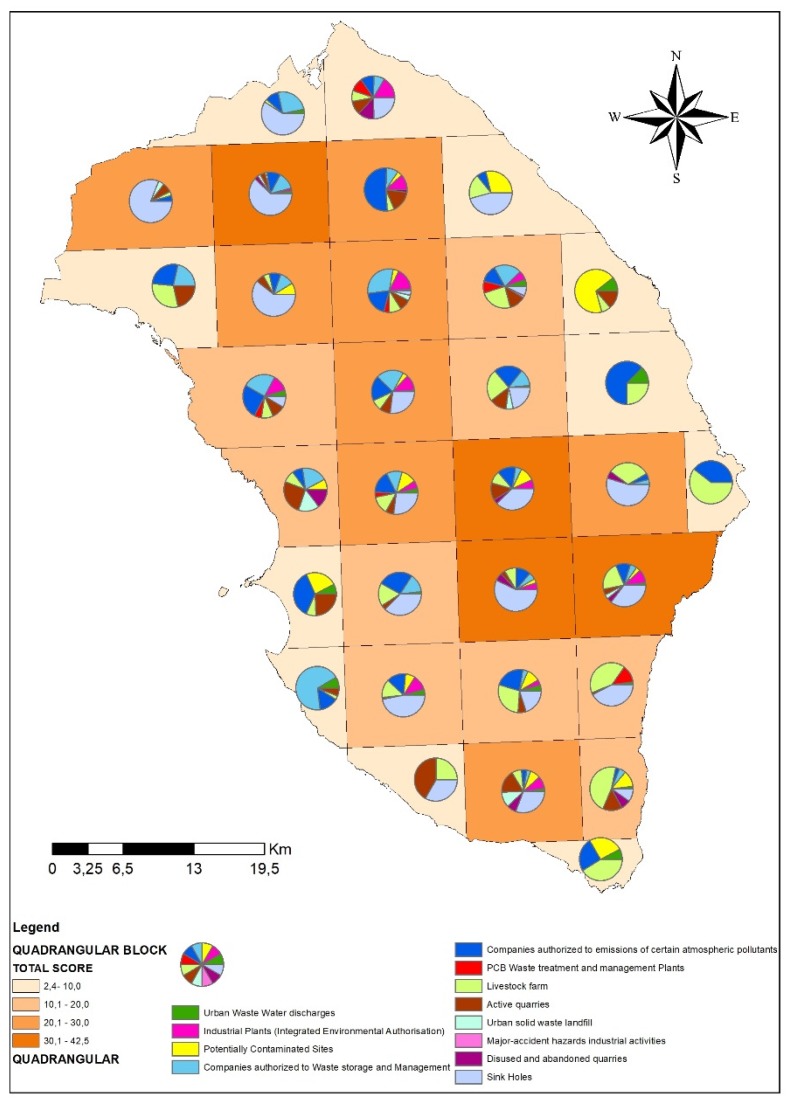
Specific contribution of each threat to groundwater quality among the 12 anthropic/environmental pressures identified in the study.

**Figure 5 ijerph-17-01835-f005:**
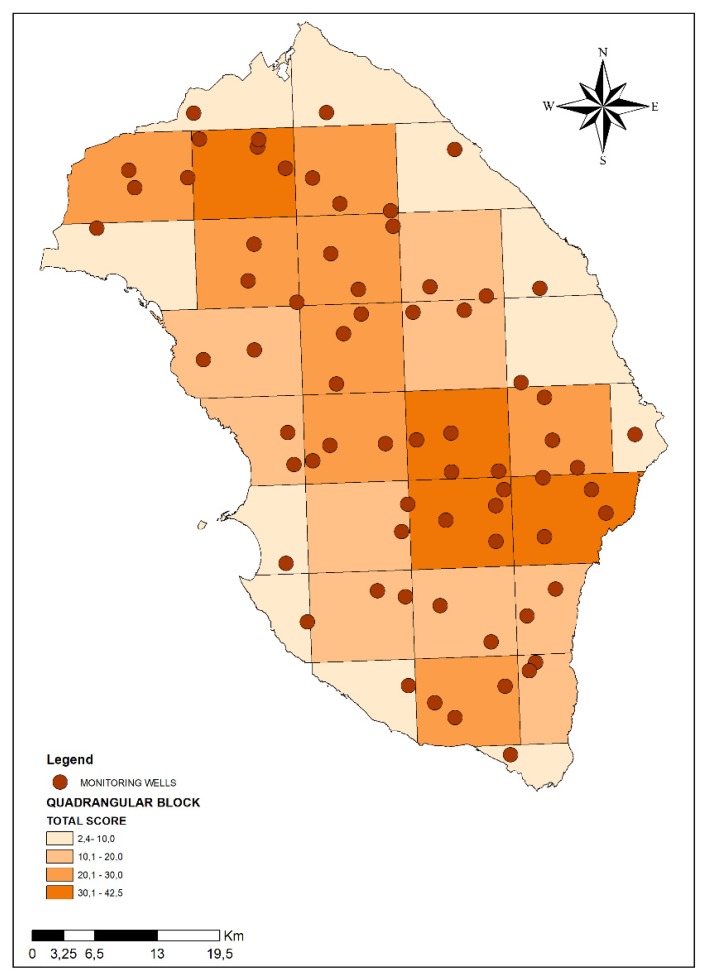
Monitoring wells identified in the 32 blocks into which Salento area has been segmented.

**Table 1 ijerph-17-01835-t001:** Score assigned to each indicator of anthropic/environmental pressure on groundwater.

	Layer	Attributed Weight
1	Urban Waste Water discharges	0,3
2	Industrial Plants (Integrated Environmental Authorization)	0,8
3	Potentially Contaminated Sites	1
4	Companies authorized for Waste storage and management	0,8
5	Companies authorized to emit certain atmospheric pollutants	0,5
6	PCB Waste treatment and management Plants	1
7	Livestock farm	0,1
8	Active quarries	0,2
9	Urban solid waste landfill	1
10	Major-accident hazards industrial activities	0,2
11	Disused and abandoned quarries	0,6
12	Sinkholes	0,8

**Table 2 ijerph-17-01835-t002:** Number of points/sources of pressures identified in Salento and their relative total score.

	Layer	Attributed Weight	N° of Identified Pressures	Total Score
1	Urban Waste Water discharges	0,3	34	10,2
2	Industrial Plants (Integrated Environmental Authorization)	0,8	37	29,6
3	Potentially Contaminated Sites	1	28	28
4	Companies authorized for Waste storage and management	0,8	69	55,2
5	Companies authorized to emit certain atmospheric pollutants	0,5	164	82
6	PCB Waste treatment and management Plants	1	7	7
7	Livestock farm	0,1	770	77
8	Active quarries	0,2	222	44,4
9	Urban solid waste landfill	1	10	10
10	Major-accident hazards industrial activities	0,2	4	0,8
11	Disused and abandoned quarries	0,6	22	13,2
12	Sinkholes	0,8	224	179,2

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
