# Peer review of "Water Quality Assessment: A Quali-Quantitative Method for Evaluation of Environmental Pressures Potentially Impacting on Groundwater, Developed under the M.I.N.O.Re. Project"

_ijerph, 2020, doi:10.3390/ijerph17061835_

Round 1

Reviewer 1 Report

The authors proposed the quali-quantitative method to assess the water quality at province of Lecce. The results showed that the proposed method can evaluate the environmental pressures suffered by aquifers and the quality of the groundwater.

However, the method of quali-quantitative is simple in this paper, the majority of the outcomes are quite obvious, thus not very interesting for a reader. Apart from this main drawback, which should require an extremely deep revision of the manuscript, there are other main issues that the authors surely must resolve before the manuscript can be suitable for publication:

The title of this paper is “Water Quality Assessment: A Quali-Quantitative……”. The conclusion is the methodology can access the environment pressures suffered by aquifers and the quality of the groundwater. Please double check the reasonable of this title. Line 124: 104 wells. Line 138:70 wells. How about the water quality of these wells? How to select the 12 indicators? Please explain clearly in the paper. Lines 246-248: How to divide into 4 classes? What is the basis? 4th class: from 30,01 to 42,50. Why the upper limit is 4250? Please add the specific calculation methods in the modeling approach. How to calculate the attributed weight? Please explain clearly in the paper. The authors are asked to clearly state which is the novelty of the paper for which the paper deserves publication.

8. What the authors are trying to achieve, and finally what are the primary conclusions to be beneficial for IJERPH? It is not clear from the conclusions.

Author Response

Dear Reviewer,

we thank you for interest in our paper and for helpful suggestions and encouragement.

We have revised the Manuscript (Number: ijerph-709777; Title: Water Quality Assessment: A Quali-Quantitative Method for evaluation of Environmental Pressures Potentially Impacting on Groundwater, developed under M.I.N.O.Re. Project) in line with your comments.

The revisions have been highlighted, using the "Track Changes" function in Microsoft Word.

We hope you find our revised manuscript acceptable for publication.

We look forward to hearing from you.

Yours sincerely,

Prisco Piscitelli

Revision notes

The title of this paper is “Water Quality Assessment: A Quali-Quantitative……”. The conclusion is the methodology can access the environment pressures suffered by aquifers and the quality of the groundwater. Please double check the reasonable of this title.

The title “Water Quality Assessment: A Quali-Quantitative Method for evaluation of Environmental Pressures Potentially Impacting on Groundwater, developed under M.I.N.O.Re. Project” was chosen because in this study has been adopted a quali-quantitative approach for the assessment of environmental pressures on groundwater as rationale to determine monitoring wells representative of the vulnerability of the aquifer in Salento area. From the qualitative point of view were chosen different type of pressures potentially impacting on the aquifer; to each kind of pressure a score has been attributed, on the basis of the potential impact on groundwater, in order to define them quantitatively.

-The paragraph “Therefore has been adopted a quali-quantitative approach for the assessment of environmental pressures on groundwater as rationale to determine monitoring wells representative of the vulnerability of the aquifer in Salento area. From the qualitative point of view were chosen different type of pressures potentially impacting on the aquifer and to each kind of pressure a score has been attributed, on the basis of the potential impact on groundwater, in order to define them quantitatively.” has been added in the conclusions (lines 447-452).

Line 124: 104 wells. Line 138:70 wells. How about the water quality of these wells?

This is primarily a methods paper, which presents no water quality results and only discusses the approach used in the second objective of the M.I.N.O.Re. project for establishing a groundwater sampling plan.

-This has been best explained by adding in the introduction, line 141, the following sentence “The second objective of the M.I.N.O.Re. project is discussed in this article.”;

-lines 159-162: The paragraph “Therefore, in this paper we describe the integrated approach used under M.I.N.O.Re. project for the assessment of groundwater quality in the Salento area, that included individuation of the main anthropic or environmental pressures and geological mapping” has been replaced with “Therefore, in this paper we describe the integrated approach used under M.I.N.O.Re. project for establishing a groundwater sampling plan in the Salento area, that included individuation of the main anthropic or environmental pressures, their weight and geological mapping.”

How to select the 12 indicators? Please explain clearly in the paper.

- Materials and Methods, lines 199-205: The paragraph “In order to define the localization of the wells to be monitored, within a maximum pre-defined number of about 70 wells, we have decided to start with the definition of a complete list of local environmental pressures and human activities (i.e. industries, legal/illegal waste disposal sites etc.) potentially impacting on groundwater. We have also taken into account the most updated knowledge and recent scientific literature concerning major hydrogeological variables (i.e. flow direction; the presence of cracks, caves, sinkholes etc.) that can potentially increase the vulnerability of groundwater in terms of pollutants’ load distribution, infiltration or diffusion [15,19,20].” has been modified in “In order to define the localization of the wells to be monitored, within a maximum pre-defined number of about 70 wells, we have decided to start with the definition of a complete list of local environmental pressures and human activities (i.e. industries, legal/illegal waste disposal sites etc.) potentially impacting on groundwater, based on the most updated knowledge and recent scientific literature [23-27]. We have also taken into account the major hydrogeological variables (i.e. flow direction; the presence of cracks, caves, sinkholes etc.) that can potentially increase the vulnerability of groundwater in terms of pollutants’ load distribution, infiltration or diffusion [19,28,29].”

Lines 246-248: How to divide into 4 classes? What is the basis? 4th class: from 30,01 to 42,50. Why the upper limit is 4250?

We decided to divide into 4 classes on the basis of total scores.

-Line 351: The text “4th class: from 30,01 to 42,50” has been modified in “4th class: more than 30,0”

Please add the specific calculation methods in the modeling approach. How to calculate the attributed weight? Please explain clearly in the paper.

-Materials and Methods, lines 289-294: The paragraph “After entering the information levels in the GIS database, the "weight" that was assigned to each individual information layer was established. Obviously, each layer had a different weight depending on its relevance in terms of possible threats and pressure on groundwater. Therefore, a score from 0 to 1 was assigned to each one of the 12 parameters/layers used as indicator of anthropic/environmental pressure on groundwater and the number of sources of anthropic/environmental pressure in the entire province of Lecce was computed thanks to information provided by the local health authority ASL Lecce and the regional environmental agency ARPA Puglia” has been modified in “After entering the information levels in the GIS database, the "weight" that was assigned to each individual information layer was established, on the basis of prior knowledge and scientific literature [23]. Then were given rating values ranging from 0 to 1 depending on the contribution of each anthropic/environmental pressure to pollution of groundwater. Obviously, each layer had a different weight depending on its relevance in terms of possible threats and pressure on aquifer (Table 1).”

-The following paragraph has been added in methods, lines 299-303:

“For each block, into which the entire province of Lecce was divided, has been calculated the total score as the weighted sum of the twelve layers using the following equation in a GIS tool:

(i=anthropic/environmental pressure; n=number of points of pressure i; w=weight attributed to pressure i).

The authors are asked to clearly state which is the novelty of the paper for which the paper deserves publication. What the authors are trying to achieve, and finally what are the primary conclusions to be beneficial for IJERPH? It is not clear from the conclusions.

-The paragraph “This manuscript presents a clear and practical approach for establishing a groundwater sampling plan. Therefore, the paper could be a useful resource for public authorities with a mandate to monitor groundwater quality.” has been added in the conclusions (lines 452-454).

The following references have been added:

  1. Kurwadkar, S. Groundwater Pollution and Vulnerability Assessment. Water Environ Res 2017, 89(10), 1561-1579.
  2. Al-Wabel, M.; El-Saeid, M.H.; El-Naggar, A.H.; Al-Romian, F.A.; Osman, K.; Elnazi, K.; Sallam, A.S. Spatial distribution of pesticide residues in the groundwater of a condensed agriculture area. Arab J Geosci 2016, 9(120), 1-10.
  3. Bhutiani, R.; Kulkarni, D.B.; Khanna, D.R.; Gautam, A. Water Quality, Pollution Source Apportionment and Health Risk Assessment of Heavy Metals in Groundwater of an Industrial Area in North India. Expo Health 2016, 8(1), 3-18.
  4. Chen, S.; Jiao, X.; Gai, N.; Li, X.; Wang, X.; Lu, G.; Piao, H.; Rao, Z.; Yang, Y. Perfluorinated compounds in soil, surface water, and groundwater from rural areas in eastern China. Environ Pollut 2016, 211, 124-131.
  5. Deshmukh, K.K.; Aher, S.P. Assessment of the impact of municipal solid waste on groundwater quality near Sangamner City using GIS approach. Water Resour Manage 2016, 30, 2425-2443.
  6. Ceplecha, Z.L.; Waskom, R.M.; Bauder, T.A.; Sharkoff, J.L.; Khosla, R. Vulnerability assessments of Colorado groundwater to nitrate contamination. Water, Air, Soil Pollut 2004, 159, 373-394.
  7. Cucchi, F.; Franceschini, G.; Zini, L.; Aurighi, M. Intrinsic vulnerability assessment of Sette Comuni plateau aquifer (Veneto region, Italy). J Environ Manag 2008, 88, 984-994.
  8. Baalousha, H. Assessment of a groundwater quality monitoring network using vulnerability mapping and geostatistics: a case study from Heretaunga Plains, New Zealand. Agric Water Manag 2010, 97(2), 240-246.
  9. Gogu, R.C.; Dassargues, A. Current trends and future challenges in groundwater vulnerability assessment using overlay and index methods. Environ Geol 2000, 39(6), 549-559.

Reviewer 2 Report

This manuscript presents a clear and practical approach for establishing a groundwater sampling plan. This is primarily a methods paper, which presents no water quality results. Nevertheless, the paper could be a useful resource for public authorities with a mandate to monitor groundwater quality and I believe that the subject is appropriate for publication in IJERPH.  Prior to publication, I recommend that the authors improve the manuscript with the following revisions:

1) In the introduction: provide a review of existing groundwater sampling methodologies, demonstrating how the approach presented here is an improvement.

2) In the methods section: provide a rigorous rationale for each aspect of the methodology. If the paper is to be a resource for other practitioners, more details are needed so help them replicate the steps. Specifically:

  • Why is the number of sampling locations limited to 70?
  • Why did the authors divide the area in 32 blocks? How was this number chosen?
  • How did the authors assign weights to each of the risk layers? Was this arbitrary? Based on prior knowledge? Have the authors considered using principal component analysis to assign weights?
  • Explain more clearly how scores were computed for each pixel. Was each risk factor turned into a binary variable, equal to 0 or 1 on each pixel, before computing the weighted average?

3) Correct the following:

  1. Typo on line 112: on these this
  2. Sentence starting on line 125 is a missing a word, perhaps “wells”.
  3. Line 138: “70 wells that should be adequately representative of the quality of the aquifer”. This sentence incorrectly describes the approach developed here, whose goal is not to be “representative” (i.e., the goal is not to get a representative average of contaminants concentrations), but rather to capture and detect all possible contaminants. Rather than being “representative”, the approach gives more weight to areas that have more contamination sources.
  4. Line 170: delete “been”.
  5. Line 170: “being as a tool”
  6. Line 195: “companies authorized to emissions of emit certain pollutants”.
  7. Lines 330-331: “has a precious importance from an environmental point of view”. I recommend replacing with “is relevant for environmental monitoring applications”.

Author Response

Dear Reviewer,

we thank you for interest in our paper and for helpful suggestions and encouragement.

We have revised the Manuscript (Number: ijerph-709777; Title: Water Quality Assessment: A Quali-Quantitative Method for evaluation of Environmental Pressures Potentially Impacting on Groundwater, developed under M.I.N.O.Re. Project) in line with your comments.

The revisions have been highlighted, using the "Track Changes" function in Microsoft Word.

We hope you find our revised manuscript acceptable for publication.

We look forward to hearing from you.

Yours sincerely,

Prisco Piscitelli

Revision notes

1) In the introduction: provide a review of existing groundwater sampling methodologies, demonstrating how the approach presented here is an improvement.

-The paragraph “The existing groundwater sampling methodologies are based only on the assessment of the intrinsic groundwater vulnerability to pollution, without considering the pressures present in the area [11-13]. Index methods are very popular in vulnerability assessment where classification of aquifer area is done based on geological and hydrogeological factors [14].” has been added in the introduction (lines 144-148).

2) In the methods section: provide a rigorous rationale for each aspect of the methodology. If the paper is to be a resource for other practitioners, more details are needed so help them replicate the steps. Specifically:

  • Why is the number of sampling locations limited to 70?

-Introduction, lines 142-143: The text “(established on the basis of the project budget)” has been added.

  • Why did the authors divide the area in 32 blocks? How was this number chosen?

The number of 32 blocks is the result of the subdivision used.

- Materials and Methods, lines 206-211: The paragraph “Therefore, in order to determine the main anthropic and natural pressures on the aquifer, we have generated a map of the entire province of Lecce using the geographic information system (GIS) and interpolated it with a grid that led to the subdivision of the study area into 32 quadrangular blocks measuring 10 km x 10 km.” has been modified in “Therefore, in order to determine the main anthropic and natural pressures on the aquifer, we have generated a map of the entire province of Lecce using the geographic information system (GIS) and interpolated it with a grid that led to the subdivision of the study area into quadrangular blocks measuring 10 km x 10 km. Along the coast, due to the presence of the sea, some blocks presented smaller dimensions, so that they were incorporated into the nearest ones. As a result of this subdivision, the study area was divided into 32 blocks.”

  • How did the authors assign weights to each of the risk layers? Was this arbitrary? Based on prior knowledge? Have the authors considered using principal component analysis to assign weights?

-Materials and Methods, lines 289-294: The paragraph “After entering the information levels in the GIS database, the "weight" that was assigned to each individual information layer was established. Obviously, each layer had a different weight depending on its relevance in terms of possible threats and pressure on groundwater. Therefore, a score from 0 to 1 was assigned to each one of the 12 parameters/layers used as indicator of anthropic/environmental pressure on groundwater and the number of sources of anthropic/environmental pressure in the entire province of Lecce was computed thanks to information provided by the local health authority ASL Lecce and the regional environmental agency ARPA Puglia” has been modified in “After entering the information levels in the GIS database, the "weight" that was assigned to each individual information layer was established, on the basis of prior knowledge and scientific literature [23]. Then were given rating values ranging from 0 to 1 depending on the contribution of each anthropic/environmental pressure to pollution of groundwater. Obviously, each layer had a different weight depending on its relevance in terms of possible threats and pressure on aquifer (Table 1).”

  • Explain more clearly how scores were computed for each pixel. Was each risk factor turned into a binary variable, equal to 0 or 1 on each pixel, before computing the weighted average?

-The table 1 has been modified and the table 2 has been added in results.

-The following paragraph has been added in methods, lines 299-303:

“For each block, into which the entire province of Lecce was divided, has been calculated the total score as the weighted sum of the twelve layers using the following equation in a GIS tool:

(i=anthropic/environmental pressure; n=number of points of pressure i; w=weight attributed to pressure i).

-The following paragraph has been added in results, lines 340-342:

“Thanks to information provided by the local health authority ASL Lecce and the regional environmental agency ARPA Puglia, the total number of the analyzed anthropic/environmental pressures, that persist on the Salento, and their relative total score have been obtained (Table 2).”

3) Correct the following:

  1. Typo on line 112: on these this
  2. Sentence starting on line 125 is a missing a word, perhaps “wells”.
  3. Line 138: “70 wells that should be adequately representative of the quality of the aquifer”. This sentence incorrectly describes the approach developed here, whose goal is not to be “representative” (i.e., the goal is not to get a representative average of contaminants concentrations), but rather to capture and detect all possible contaminants. Rather than being “representative”, the approach gives more weight to areas that have more contamination sources.
  4. Line 170: delete “been”.
  5. Line 170: “being as a tool”
  6. Line 195: “companies authorized to emissions of emit certain pollutants”.
  7. Lines 330-331: “has a precious importance from an environmental point of view”. I recommend replacing with “is relevant for environmental monitoring applications”.

All the above corrections have been made.

The following references have been added:

  1. Kurwadkar, S. Groundwater Pollution and Vulnerability Assessment. Water Environ Res 2017, 89(10), 1561-1579.
  2. Al-Wabel, M.; El-Saeid, M.H.; El-Naggar, A.H.; Al-Romian, F.A.; Osman, K.; Elnazi, K.; Sallam, A.S. Spatial distribution of pesticide residues in the groundwater of a condensed agriculture area. Arab J Geosci 2016, 9(120), 1-10.
  3. Bhutiani, R.; Kulkarni, D.B.; Khanna, D.R.; Gautam, A. Water Quality, Pollution Source Apportionment and Health Risk Assessment of Heavy Metals in Groundwater of an Industrial Area in North India. Expo Health 2016, 8(1), 3-18.
  4. Chen, S.; Jiao, X.; Gai, N.; Li, X.; Wang, X.; Lu, G.; Piao, H.; Rao, Z.; Yang, Y. Perfluorinated compounds in soil, surface water, and groundwater from rural areas in eastern China. Environ Pollut 2016, 211, 124-131.
  5. Deshmukh, K.K.; Aher, S.P. Assessment of the impact of municipal solid waste on groundwater quality near Sangamner City using GIS approach. Water Resour Manage 2016, 30, 2425-2443.
  6. Ceplecha, Z.L.; Waskom, R.M.; Bauder, T.A.; Sharkoff, J.L.; Khosla, R. Vulnerability assessments of Colorado groundwater to nitrate contamination. Water, Air, Soil Pollut 2004, 159, 373-394.
  7. Cucchi, F.; Franceschini, G.; Zini, L.; Aurighi, M. Intrinsic vulnerability assessment of Sette Comuni plateau aquifer (Veneto region, Italy). J Environ Manag 2008, 88, 984-994.
  8. Baalousha, H. Assessment of a groundwater quality monitoring network using vulnerability mapping and geostatistics: a case study from Heretaunga Plains, New Zealand. Agric Water Manag 2010, 97(2), 240-246.
  9. Gogu, R.C.; Dassargues, A. Current trends and future challenges in groundwater vulnerability assessment using overlay and index methods. Environ Geol 2000, 39(6), 549-559.

Round 2

Reviewer 1 Report

Paper has been revised adequately, hence it's now in an acceptable stage.

Reviewer 2 Report

The authors have adequately addressed my comments. I recommend doing a final check on the English and grammar. For example, the new sentence starting on line 478 is lacking a subject.